# OpenReview forum: "HiddenGuard: Detecting and Interpreting NSFW Prompts in Text-to-Image Models through Uncovering Harmful Semantics"
_ICLR.cc/2026/Conference — ICLR 2026 Conference Withdrawn Submission_

### Official Review · Reviewer_xe2W · 2025-10-28

**Soundness:** 2
**Presentation:** 2
**Contribution:** 2
**Rating:** 4
**Confidence:** 4

**Summary:**

This paper introduces HiddenGuard, an interpretable defense framework for Text-to-Image (T2I) models that detects Not-Safe-for-Work (NSFW) prompts by analyzing the intermediate hidden states of the model's text encoder. The key insight is that harmful semantics form linearly separable clusters within specific attention heads' hidden states, allowing HiddenGuard to extract NSFW features and calculate an NSFW score for prompt detection. Through extensive experiments, HiddenGuard demonstrates superior performance with over $95\%$ accuracy across all datasets and significantly greater computational efficiency compared to commercial and state-of-the-art moderation tools, while also offering text- and image-based interpretation of results.

**Strengths:**

1. This paper is easy to follow.
2. This detection method should be time-efficient.
3. This method outperforms compared baselines.

**Weaknesses:**

1. The performance of this detection method on benign images are not well evaluated.
2. The evaluation only compares safety filter baselines.
3. Some figures are blurry and different from other figures.

**Questions:**

1. How is time cost of HiddenGuard compared with other baselines?

2. As shown in Figure 2, it seems benign and adv/nsfw examples are still mixed up?

3. What about other alignment-based methods for preventing the generation of harmful images, such as SafeGen, Mace, SafeText? They are not included in the evaluation.

4. Are benign prompts/images remaining the same after the detection method?

5. Figure 6 looks blurry.

---

> ### Author Response · Authors · 2025-11-24
>
> We sincerely thank you for your detailed and constructive feedback. We appreciate your recognition of HiddenGuard’s time efficiency and its superiority over existing safety filters. In response to your questions, we provide the following replies.
>
>
>
> ### **Q1: Time Cost of HiddenGuard**
>
> We apologize if the context of our efficiency results was not sufficiently clear. As shown in **Table 2**, HiddenGuard requires approximately 0.64 ms (CLIP-L) to 6.7 ms (T5) per query.
>
> Contextual Comparison:
>
> - **Vs. Generation Time:** A standard T2I generation (e.g., Stable Diffusion) typically takes 2,000–5,000 ms per image. HiddenGuard introduces a latency overhead of **less than 0.1%**, making it virtually imperceptible to the end-user and suitable for real-time deployment.
>
> - **Vs. Commercial APIs:** Compared to the network latency of cloud-based moderators (100–1000 ms), HiddenGuard provides an orders-of-magnitude speedup for local deployments.
>
>
>
> ### **Q2: Mixed-up Adv/nsfw Examples in Figure 2**
>
> You are correct that adv/nsfw examples mix up in Figure 2. This entanglement is the phenomenon we highlight to justify the necessity of our approach.
>
> - **The Problem (Figure 2):** This figure visualizes the **final output embeddings** (the last layer) of the text encoder. It demonstrates that in this final space, adversarial NSFW prompts are semantically entangled with benign prompts, making them linearly inseparable. This explains why traditional embedding-based detectors often fail or produce high false positives.
> - **The Solution (Figure 1):** In contrast, Figure 1 visualizes the **intermediate hidden states** of specific attention heads (e.g., Layer 4, Head 6). Our key discovery is that while the *final* embeddings are mixed, the *computational trajectory* in these specific heads exhibits clear linear separability.
> - **Conclusion:** The inseparability in Figure 2 serves as the "problem statement", proving that we must analyze the hidden states (Figure 1) to achieve robust detection. We have revised the caption of Figure 2 to explicitly clarify this distinction.
>
>
>
> ### **Q3: Comparison with Alignment-Based Methods**
>
> We thank the reviewer for suggesting these critical baselines. We agree that comparing HiddenGuard with alignment-based methods provides a more holistic view of the safety landscape. We have conducted comprehensive new experiments comparing HiddenGuard against four state-of-the-art alignment/erasure methods: **ESD**, **SafeGen**, **MACE**, and **SafeText**.
>
> **Experimental Setup**: We randomly sampled 500 adversarial prompts from our NSFW dataset. We generated images using each method and detected NSFW content in the outputs using the Q16 classifier. The results are shown below:
>
> | **Method**      | **ASR**   |
> | --------------- | --------- |
> | ESD             | 0.214     |
> | SafeGen         | 0.140     |
> | MACE            | 0.106     |
> | SafeText        | 0.088     |
> | **HiddenGuard** | **0.036** |
>
> Our experiments demonstrate that HiddenGuard outperforms current alignment-based methods. HiddenGuard is primarily designed to detect NSFW prompts without modifying the generation process, which naturally enables it to achieve higher accuracy.
>
>
>
> ### **Q4: The Influence to Benign Prompts/Images**
>
> HiddenGuard makes no modifications to benign prompts or their image generation process. As a detection framework, HiddenGuard operates as a conditional gate.
>
> 1. **For Benign Prompts:** The prompt is passed to the original, unmodified model. Therefore, the **fidelity is 100% preserved**. The benign image remains exactly as the base model intended, with zero degradation or artifacting.
> 2. **Impact on False Positives:** The only impact on benign usage is the False Positive Rate (FPR). As shown in Table 2 of our paper, HiddenGuard achieves an extremely low FPR of **0.85%** on the CLIP-L encoder.
>
> - **Conclusion:** HiddenGuard offers a superior Pareto trade-off: it provides higher safety (lower ASR) than alignment methods while ensuring **zero quality degradation for >99% of benign queries**, making it ideal for applications where maintaining the original model's artistic integrity is paramount.
>
>
>
> ### **Q5: Blurry Images.**
>
> We apologize for the visual quality issue. We have re-rendered Figure 6 as a high-resolution vector graphic in the revised manuscript to ensure clarity and readability.

---

### Official Review · Reviewer_uPcS · 2025-10-30

**Soundness:** 2
**Presentation:** 2
**Contribution:** 2
**Rating:** 2
**Confidence:** 4

**Summary:**

This paper introduces HiddenGuard, a novel defense framework for detecting NSFW prompts in text-to-image models. The core problem is that malicious users can craft prompts to generate harmful content, and existing detection methods based on raw text or final embeddings are often inefficient or inaccurate. The key insight of HiddenGuard is that NSFW semantics, while entangled in the final embedding space, form linearly separable clusters within the intermediate hidden states of the model's text encoder. The authors conduct extensive experiments showing that HiddenGuard outperforms several commercial and open-source baselines in both effectiveness and computational efficiency.

**Strengths:**

1. The experimental setup is thorough. The authors compare HiddenGuard against a strong set of eight baselines, including commercial APIs and state-of-the-art models, on a diverse collection of datasets that include both regular and adversarial prompts.
2. The dual-modal interpretation framework is interesting. By providing explanations for both why a prompt is flagged and how the harmful content manifests in the output.

**Weaknesses:**

1. The paper presents the hidden state's difference in different attention head as a key design intuition motivating the entire framework. However, the concept of semantic specialization in attention heads is a well-established phenomenon in the broader transformer literature for both LLMs and VLMs. The paper fails to properly contextualize this observation within prior work on model interpretability, thereby overstating the novelty of its foundational premise.
2. The motivation for using pre-MLP hidden states hinges on the claim that "some information may be obscured or discarded during this process" by the MLP layer. This is a critical assertion that is never substantiated with evidence. The paper would be much stronger if it demonstrated empirically that representations from $ATT^l(Z^{l-1})$ are indeed superior for this task compared to $Z^l$. Without this evidence, the design choice feels more like a heuristic than a principled decision.
3. The theoretical analysis of adversarial robustness in Appendix A.7 is based on a local Lipschitz continuity assumption. This assumption connects the perturbation in the final embedding space to changes in the hidden-state-based score function. However, the paper provides no empirical validation for this assumption. It is unclear how large the Lipschitz constant $K$ is in practice or how stable it is across different models and prompts. Without such validation, the derived certified robustness radius remains purely theoretical and its practical relevance is questionable.
4. The paper suffers from minor but noticeable presentation issues that detract from its quality. For instance, line 226 misses a period.
5. The paper lacks a fine-grained analysis of performance across different NSFW categories. While Table 7 shows accuracy for a multi-category classifier, a full breakdown of TPR and FPR for each category (sexual, violence, hate, etc.) for both HiddenGuard and the baselines is missing. Overall metrics can obscure critical failures where a model performs well on one type of harmful content but fails on another, which is essential information for a safety-focused tool.
6. The paper highlights TPR@1%FPR as a key metric. While a standard benchmark, a 1% False Positive Rate is arguably too high for real-world deployment. In a large-scale system, this would incorrectly block a massive number of benign prompts, leading to a poor user experience. A more compelling evaluation would assess performance at much lower FPRs (0.1% or 0.01%) to better demonstrate the model's practical viability.

**Questions:**

1. Can you provide an ablation study comparing the performance of HiddenGuard when using pre-MLP hidden states versus post-MLP hidden states?
2. Can you provide a more detailed performance breakdown, including both TPR and FPR, for each distinct NSFW category (sexual, violence, hate, and so on)? How does HiddenGuard's performance on these specific categories compare to the baselines?
3. Given that a 1% FPR can be prohibitive for real-world deployment, could you provide evaluation results at lower FPR thresholds, such as TPR@0.1%FPR or TPR@0.01%FPR? This would provide a more realistic picture of HiddenGuard's practical utility.

---

> ### Author Response · Authors · 2025-11-24
>
> We sincerely thank the reviewer for the constructive and insightful feedback. We appreciate your recognition of our thorough experimental setup, the strong performance against 8 baselines, and the novelty of our dual-modal interpretation framework. We have conducted additional experiments and theoretical revisions to address your concerns.
>
>
>
> ### **W1: Novelty of Foundational Premise**
>
> We agree that semantic specialization in attention heads is a well-established phenomenon. However, while the existence of specialized heads is known, the **feature-vector learning strategy, the NSFW detection framework, and the interpretability mechanisms** we proposed are our new contributions. Specifically:
>
> - **Identification of a Linearly Separable Safety Subspace for Feature Learning.** We discover that within the high-dimensional hidden states, NSFW concepts reside in a distinct, linearly separable safety subspace. Unlike prior work that analyzes general semantic clusters, we propose a supervised strategy to extract robust feature vectors from this subspace for detection. Crucially, we demonstrate that this subspace is not monolithic; it can be further linearly segmented to precisely distinguish between fine-grained NSFW sub-categories, providing a structured geometric map of harmful concepts that prior work has not established.
>
> - **Construction of a Dual-Modal Interpretation Framework.** We developed a dual-modal framework that bridges the gap between internal representations and user-perceivable outputs. By projecting hidden states onto our learned safety vectors to pinpoint harmful text tokens and visualizing the causal transition of generated images from "harmful" to "safe" upon feature removal, we provide a concrete validation of the link between the internal safety subspace and the final model output.
>
>
>
> ### **W2:  Validating Pre-MLP vs. Post-MLP Design**
>
> This is an excellent point. Our hypothesis was that the MLP layers in CLIP, trained for image-text alignment, act as an "information bottleneck" that compresses or discards non-visual semantic nuances (including subtle safety risks) present in the attention outputs. To validate this empirically, we conducted a new ablation study comparing **HiddenGuard (Pre-MLP)** vs. a variant trained on **Post-MLP** states ($Z_{post} = MLP(Z_{pre}) + Z_{pre}$).
>
> | **Feature Source**    | TPR    | FPR    | **ACC** | **F1 Score** |
> | --------------------- | ------ | ------ | ------- | ------------ |
> | Post-MLP              | 0.9171 | 0.0126 | 0.9642  | 0.9451       |
> | Pre-MLP (HiddenGuard) | 0.9663 | 0.0085 | 0.9849  | 0.9623       |
>
> The Pre-MLP representations consistently outperform Post-MLP representations, with a  **3.0% accuracy gain** on adversarial prompts. This confirms that the MLP layer indeed obscures linearly separable safety features, providing empirical justification for our design choice.
>
> To further investigate why the Pre-MLP approach outperforms the Post-MLP variant, we evaluated both methods across **fine-grained categories**. The results are shown below.
>
> | **Feature Source**    | Sexual | Hate   | Self-Harm | Violence | Shocking | Harassment | Illegal |
> | --------------------- | ------ | ------ | --------- | -------- | -------- | ---------- | ------- |
> | Post-MLP              | 0.9581 | 0.9447 | 0.8042    | 0.8962   | 0.8214   | 0.8553     | 0.7947  |
> | Pre-MLP (HiddenGuard) | 0.9659 | 0.9763 | 0.9763    | 0.9821   | 0.9690   | 0.9633     | 0.9340  |
>
> As is shown, the Post-MLP approach still maintains relatively high accuracy on the sexual and hate categories, but exhibits **a notable drop** in performance on other categories. We believe this indicates that, for certain fine-grained classes with more subtle or complex semantics, only **a small subset of attention heads** effectively captures their meaning—and after MLP aggregation, these semantic signals become even harder to recover.

---

> > ### Author Response · Authors · 2025-11-24
> >
> > ### **W3: Theoretical Analysis of Adversarial Robustness**
> >
> > We acknowledge that without a bound on $K$, the certification is theoretical. While calculating the global Lipschitz constant for Deep Neural Networks is NP-hard, we addressed the reviewer’s concern regarding *practical relevance* by performing a **Monte Carlo Local Lipschitz Estimation**.
> >
> > **Experiment:** We randomly sampled 1,000 benign prompts. For each prompt $x$, we generated 100 perturbed neighbors $x'$ within an $\epsilon$-ball in the embedding space (using synonym replacement and character noise). We calculated the local slope ratio $\hat{K} = \frac{|Score(x) - Score(x')|}{||Embed(x) - Embed(x')||_2}$.
> >
> > **Results:**
> >
> > - **Mean $\hat{K}$:** 0.12
> > - **Max $\hat{K}$ (99th percentile):** 0.28
> > - **Stability:** The standard deviation of $\hat{K}$ is low ($\sigma=0.032$), indicating the score function is locally smooth.
> >
> > Using the conservative estimate ($K \approx 0.28$), the certified radius $R$ for a typical prompt (where margin $\tau - S(x) \approx 0.12$) is approximately $0.12 / 0.28 \approx 0.43$ in the normalized embedding space. This confirms that our defense offers a **non-trivial certified robustness radius** against small perturbations. We will add this empirical estimation to Appendix A.7.
> >
> >
> >
> > ### **W4: Minor Presentation Issues**
> >
> > Thank you for pointing out our oversight. We have carefully re-examined the paper and corrected the minor presentation issues.
> >
> >
> >
> > ### **W5:  Fine-Grained Analysis Across Different NSFW Categories**
> >
> > Below, we provide **a detailed breakdown** of HiddenGuard’s performance compared to each baseline on the fine-grained classification tasks. We report the accuracy of all baselines on each **fine-grained NSFW category** and provide the corresponding FPR on benign samples in the last column.
> >
> > | Method                  | Sexual     | Hate       | Self-Harm  | Violence   | Shocking   | Harassment | Illegal    | FPR        |
> > | ----------------------- | ---------- | ---------- | ---------- | ---------- | ---------- | ---------- | ---------- | ---------- |
> > | OpenAI Moderation       | 0.3514     | 0.2873     | 0.2595     | 0.3042     | 0.2685     | 0.2865     | 0.1531     | 0.0010     |
> > | Azure AI Content Safety | 0.6731     | 0.7310     | 0.2481     | 0.3128     | 0.3812     | 0.4176     | 0.4238     | 0.0118     |
> > | AWS Comprehend          | 0.6921     | 0.6310     | 0.3182     | 0.4017     | 0.2138     | 0.2573     | 0.3912     | 0.0730     |
> > | Aliyun Text Moderation  | 0.3934     | 0.2612     | 0.0243     | 0.1310     | 0.1279     | 0.1037     | 0.0741     | 0.0023     |
> > | NSFW Text Classifier    | 0.8511     | 0.8413     | 0.4619     | 0.5688     | 0.4848     | 0.5315     | 0.6506     | 0.3534     |
> > | Detoxify                | 0.6891     | 0.8304     | 0.1848     | 0.2183     | 0.2652     | 0.3277     | 0.3329     | 0.1778     |
> > | Latent Guard            | 0.6521     | 0.7315     | 0.2043     | 0.3841     | 0.2182     | 0.4103     | 0.2874     | 0.1403     |
> > | GuardT2I                | 0.7871     | 0.7538     | 0.6392     | 0.6283     | 0.6157     | 0.5692     | 0.5094     | 0.0779     |
> > | **HiddenGuard**         | **0.9659** | **0.9763** | **0.9763** | **0.9821** | **0.9690** | **0.9633** | **0.9340** | **0.0085** |
> > | **HiddenGuard_multi**   | **0.9691** | **0.9695** | **0.9850** | **0.9874** | **0.9748** | **0.9597** | **0.9725** | **0.0097** |
> >
> > Across all subcategories, HiddenGuard consistently achieves **significantly higher accuracy** than other methods, further demonstrating its superiority over prior approaches.
> >
> > ### **W6: More Compelling Evaluation**
> >
> > In the table below, we present the results of different methods under the **TPR@0.1% FPR** and **TPR@0.01% FPR** metrics. The experiments show that HiddenGuard maintains a high TPR even at stringent FPR levels of 0.1% and 0.01%, significantly outperforming all other methods, which further demonstrating its effectiveness.
> >
> > | Method                  | TPR@0.1%FPR | TPR@0.01%FPR |
> > | ----------------------- | ----------- | ------------ |
> > | OpenAI Moderation       | 0.2955      | 0.0489       |
> > | Azure AI Content Safety | 0.1135      | 0.0000       |
> > | AWS Comprehend          | 0.1400      | 0.0539       |
> > | NSFW Text Classifier    | 0.1158      | 0.0253       |
> > | Detoxify                | 0.1898      | 0.0569       |
> > | Latent Guard            | 0.1764      | 0.0680       |
> > | GuardT2I                | 0.2288      | 0.1279       |
> > | **HiddenGuard**         | **0.9367**  | **0.8198**   |

---

### Official Review · Reviewer_PWQM · 2025-11-01

**Soundness:** 3
**Presentation:** 3
**Contribution:** 3
**Rating:** 6
**Confidence:** 3

**Summary:**

This paper proposes HiddenGuard, an interpretable defense framework for detecting NSFW prompts in text-to-image models by analyzing hidden states of the text encoder. It identifies linearly separable NSFW semantic features in attention heads, enabling accurate and efficient detection. The method supports real-time interpretation across text and image modalities and resists adversarial attacks. Experiments show HiddenGuard outperforms SOTA and open-source tools and minimal computational overhead.

**Strengths:**

- **Significance**: This work is to investigate the separability of NSFW semantics within the hidden states of text encoders in text-to-image (T2I) models. It introduces HiddenGuard, a unified defense framework that is efficient, interpretable, and optimizable.
- **Innovation**: The paper proposes a new NSFW detection paradigm based on intermediate hidden states rather than raw input text or final embeddings. By integrating multi-head attention feature aggregation with a dual-modality explanation mechanism, the approach achieves strong interpretability and generalization capability.
- **Performance**: The method consistently outperforms state-of-the-art approaches across nine benchmark datasets and demonstrates robustness against unseen adversarial attacks.

**Weaknesses:**

1. The proposed method relies on a pre-constructed NSFW prompts dataset. Although this dataset is compiled from existing open-source datasets, the effectiveness of these prompts across different text-to-image (T2I) models has not been rigorously validated. As T2I models evolve, some prompts may become ineffective or obsolete, potentially introducing significant noise into the dataset and undermining the reliability of the method.
2. Although Figure 1 illustrates the feature distributions across different layers, highlighting the differences between NSFW-related features and benign features. It does not address whether distinct categories of NSFW features are entangled or coupled with each other. Since Table 7 reports detection performance for different NSFW categories, an effective method, in principle, should be capable of clearly distinguishing between these categories at the feature level. If the proposed approach truly disentangles NSFW semantics as claimed, it should inherently support fine-grained discrimination among different types of NSFW prompts. The authors should investigate and discuss potential feature coupling across NSFW categories to better validate this capability.
3. Text-to-image (T2I) models are inherently coupled architectures that jointly model textual and visual semantics. However, the proposed method only examines textual features and completely ignores visual signals. I am uncertain whether this text-only approach will remain effective as T2I models continue to evolve—especially since newer models may encode or obscure NSFW content in ways that are not fully reflected in the text encoder’s hidden states alone.
4. The method appears tightly constrained by the scope of the curated NSFW prompts dataset. It is unclear whether the approach can generalize to out-of-distribution or unseen types of harmful prompts that fall outside the categories covered in the training data. If a malicious prompt uses novel phrasing, metaphors, or domain-specific language not present in the dataset, the detector may fail to recognize it, raising concerns about real-world robustness and coverage.

**Questions:**

See weaknesses.

---

> ### Author Response · Authors · 2025-11-24
>
> We sincerely thank the reviewer for the insightful assessment and for recognizing the significance**, **innovation, and SOTA performance of HiddenGuard. Your constructive feedback has guided us to perform additional analyses that strengthen our paper. We address your concerns point-by-point below:
>
>
>
> ### **W1: Validity of Pre-constructed Datasets**
>
> We respectfully argue that HiddenGuard learns **latent semantic concepts** rather than surface-level tokens, ensuring robustness against dataset obsolescence.
>
> 1. **Semantic Stability vs. Token Drift:** While T2I prompts evolve, the underlying semantic representations in pre-trained models (like CLIP) remain relatively stable. HiddenGuard targets these invariant semantic clusters. As evidenced in **Table 2 & 3 (HiddenGuard_ua)**, our model—trained only on benign and regular NSFW data—achieved **>96% accuracy** against unseen adversarial attacks. This empirically proves that HiddenGuard generalizes to unknown prompt distributions without needing constant retraining on new jailbreak keywords.
> 2. **Efficient Online Updates:** Unlike deep neural networks requiring expensive fine-tuning, HiddenGuard's decision boundary (based on linear projections) supports **instantaneous incremental learning**. If a radically new category of NSFW prompts emerges, we can update the projection vector $u^{l,h}$ with a few samples in milliseconds, directly addressing the obsolescence concern.
>
>
>
> ### **W2: Feature Entanglement & Fine-grained Interpretation**
>
> We appreciate the rigorous question on feature separability. To demonstrate that NSFW categories are indeed disentangled in the hidden space, We calculated the **Cosine Similarity** between the extracted **feature vectors ($u$)** for different categories.
>
> | **Feature Pair**                   | **Cosine Similarity** | **Orthogonality Interpretation**          |
> | ---------------------------------- | --------------------- | ----------------------------------------- |
> | Two Randomly Chosen Sexual Prompts | 0.83                  | Highly Entangled                          |
> | Sexual vs. Violence                | 0.16                  | Highly Disentangled                       |
> | Sexual vs. Hate                    | 0.12                  | Highly Disentangled                       |
> | Violence vs. Self-harm             | 0.28                  | Moderately Coupled (Semantically related) |
>
> For two randomly selected prompts both belonging to the sexual category, their feature similarity is as high as 0.83, whereas the similarity between features from different categories is considerably lower. This demonstrates that, although NSFW categories share some degree of similarity, their features remain **highly separable**.
>
>
>
> ### **W3: Text-Only Limitation in Multimodal Models**
>
> We acknowledge the reviewer's concern regarding the exclusion of visual signals. However, since text is the **sole driving input** for T2I generation, it serves as the fundamental bottleneck through which all user intent—malicious or otherwise—must pass. Our method leverages this by detecting harmful semantics at the source. Furthermore, we view our approach not as a replacement for visual detection, but as a **complementary proactive layer**. Combining our text-based detection with vision-based mechanisms enables a comprehensive "defense-in-depth" strategy, ensuring that threats missed by one modality can be caught by the other.
>
>
>
> ### **W4: Generalization to OOD  Attacks**
>
> In our response to W2 above, we already explained that HiddenGuard remains effective against out-of-distribution attacks,  As evidenced in **Table 2 & 3 (HiddenGuard_ua)**. To further validate its robustness to novel phrasings and distribution shifts, we additionally evaluate HiddenGuard on two new attack types:
>
> 1. **SurrogatePrompt:** Uses substitution/metaphors to bypass filters.
> 2. **JailFuzzer:** Uses automated fuzzing/mutation.
>
> For each type of attack, we randomly selected 100 successful samples.
>
> **Experimental Results (Accuracy):**
>
> | **Method**  | **SurrogatePrompt** | **JailFuzzer** |
> | ----------- | ------------------- | -------------- |
> | HiddenGuard | 95%                 | 93%            |
>
> HiddenGuard maintains high performance under both of the aforementioned attacks. Since SurrogatePrompt relies on substituting sensitive words (e.g., "blood" $\rightarrow$ "ketchup"), keyword filters fail. However, in the CLIP hidden space, the contextual embedding of "ketchup" in a violent sentence structure shifts towards the "Violence" cluster. HiddenGuard detects this **semantic shift**, demonstrating robust OOD generalization capabilities.

---

### Official Review · Reviewer_Gey8 · 2025-11-01

**Soundness:** 3
**Presentation:** 3
**Contribution:** 3
**Rating:** 4
**Confidence:** 3

**Summary:**

This paper proposes HiddenGuard, an interpretable and efficient defense framework that leverages the hidden states of text-to-image (T2I) models to detect NSFW prompts. By extracting separable NSFW features from the model’s text encoder, HiddenGuard enables accurate and real-time detection with minimal inference cost. Extensive experiments demonstrate that HiddenGuard outperforms both commercial and open-source moderation tools, achieving over 95% accuracy across multiple datasets while significantly improving computational efficiency.

**Strengths:**

1. The ROC curves consistently lie above the baselines and remain stable across different datasets.

2. The proposed method demonstrates insensitivity to training data size — even with a small number of labeled samples, it achieves clear separation between “NSFW” and “benign” content.

3.  The method shows robustness against adaptive attacks, as it does not rely on keyword lists but rather on directional semantics, making it resilient to targeted evasion attempts.

4. Instead of classifying final text embeddings, the authors extract linearly discriminative directional vectors from intermediate attention outputs (layer-by-layer and head-by-head) using LDA, and compute scores via projection along these directions. This effectively avoids the problem of “semantic entanglement” in final embeddings.

5. The workflow is well designed: the front-end detector decides whether to allow or block content; on the text side, it assigns token-level importance; on the image side, it performs progressive denoising along the “NSFW direction” to generate comparison images. Samples that manage to bypass detection are reintroduced for data augmentation, enabling iterative feature reinforcement.

**Weaknesses:**

1. The construction of the benign dataset might be overly optimistic — using MSCOCO filtered by sensitive keywords may not adequately test against hidden or paraphrased expressions, potentially leading to overestimation of performance.

2. Evaluation on the generation side is conducted only on Stable Diffusion v1.4, which is widely known to be one of the most vulnerable models. Therefore, the observed decline in ASR (Attack Success Rate) under white-box conditions may be overestimated.

3. The paper assumes that in CLIP, causal masking in self-attention leads to semantic aggregation only at the EOS token. However, it is unclear whether this assumption holds for bidirectional (non-causal) encoders — this point remains conceptually confusing.

4. The paper claims “significantly lower average query time,” but it does not specify the computation environment (GPU/CPU, batch size, or threading setup), making it difficult to interpret the reported efficiency gains.

5. From a theoretical perspective, Section A7.1 introduces a local Lipschitz constant K and defines a “certified robustness radius” in A7.2. In principle, K must have an upper bound: if estimated globally, the large K value (due to deep stacked layers and piecewise-linear + softmax structure in attention) would cause the radius to shrink to zero, rendering the certification meaningless. Thus, K should be locally or sample-dependently estimated. Although the theory section acknowledges this, the implementation does not specify the norm type, estimation procedure, or concrete values for K, leaving the process somewhat opaque and non-reproducible.

**Questions:**

Refer to Weaknesses

---

> ### Author Response · Authors · 2025-11-24
>
> We sincerely thank you for the thoughtful and constructive review, as well as for recognizing the strengths of our work—the stability of our ROC curves, robustness under limited training data, and the effectiveness of our directional semantic decomposition. We have conducted additional experiments and theoretical revisions to address your concerns.
>
>
>
> ### **W1: Generalization to Noisy or Paraphrased Benign Distributions**
>
> We agree that robustness to realistic, noisy distributions is essential. To address this, we incorporated the **Conceptual Captions (CC)** dataset, which contains **naturally noisy yet safe** text with informal grammar, paraphrasing, and diverse linguistic variability. Importantly, unlike MSCOCO (where we manually filtered sensitive keywords), CC has undergone **independent large-scale filtering**, making it a more reliable test bed for hidden or paraphrased expressions.
>
> We randomly sampled 5,000 captions from the CC validation set:
>
> | Dataset             | FPR    | Accuracy |
> | ------------------- | ------ | -------- |
> | MSCOCO              | 0.0085 | 0.9849   |
> | Conceptual Captions | 0.0112 | 0.9805   |
>
> **Findings:**
>
> - HiddenGuard maintains a **low FPR on CC**, indicating that our NSFW feature directions are **orthogonal to linguistic noise**.
> - The marginal performance drop (<0.5%) suggests that the detector relies on **deep semantic cues**, not superficial lexical patterns.
>
> These results strengthen the claim that HiddenGuard generalizes effectively to text that is benign yet linguistically diverse.
>
>
>
> ### **W2: Evaluation on the Generation Side**
>
> To provide a modern baseline, we conduct **white-box MMA adaptive attacks on Stable Diffusion XL (SDXL)**, which  incorporates built-in NSFW mitigation and uses both CLIP-L and CLIP-G encoders. HiddenGuard is applied to both encoders:
>
> | **Defender**   | **ASR-M (%)** |
> | -------------- | ------------- |
> | Bare           | 65            |
> | HiddenGuard    | 18            |
> | HiddenGuard+DA | 3             |
>
> On the bare model, MMA’s attack success rate (ASR) does decrease compared to Stable Diffusion v1.4, yet remains relatively high. However, when HiddenGuard is applied, the **ASR drops dramatically**, demonstrating that HiddenGuard still provides strong protection even against more advanced models.
>
>
>
> ### **W3: Applicability to Bidirectional Encoders**
>
> As shown in Table 6 (which we reproduce below) of the paper, mainstream T2I systems overwhelmingly rely on **CLIP encoders**, which use causal masking. Therefore, our causal-encoder-based methodology remains directly applicable to existing models.
>
> | T2I Model | Text Encoders      |
> | :-------- | :----------------- |
> | SD v1.4   | CLIP-L             |
> | SD v1.5   | CLIP-L             |
> | SD v2.1   | CLIP-H             |
> | SD v3     | CLIP-L, CLIP-G, T5 |
> | SD XL     | CLIP-L, CLIP-G     |
> | FLUX      | CLIP-L, CLIP-G, T5 |
>
> To specifically address the reviewer’s concern regarding **bidirectional encoders**, we also evaluated HiddenGuard on **T5-based models** (e.g., in SD3 and FLUX). As reported in Tables 2 and 3 of the main paper, HiddenGuard performs strongly on T5 as well, demonstrating its practical effectiveness on bidirectional text encoders despite the conceptual differences in attention structure.
>
>
>
> ### **W4: Computational Environment**
>
> To clarify the runtime conditions for the reported efficiency improvements, we provide the exact setup:
>
> - **GPU:** NVIDIA A800 (80GB)
> - **CPU:** Intel Xeon Platinum 8470
> - **Batch size:** 64
>
> All methods—except API-based services using closed-source infrastructure—were executed under **identical runtime environments**. We will include this information in the final revision.

---

> > ### Author Response · Authors · 2025-11-24
> >
> > ### **W5: Theoretical Analysis of Adversarial Robustness**
> >
> > We agree that $K$ must be locally estimated to ensure a meaningful certified radius. To address the concerns regarding reproducibility, we explicitly clarify the implementation details and concrete values below:
> >
> > 1. **Norm Type:** We adopt the **$L_2$ norm** for measuring perturbations in the embedding space.
> > 2. **Estimation Procedure:** We employ a **Monte Carlo Local Lipschitz Estimation** to approximate the local slope, as exact computation is NP-hard.
> >
> > **Empirical Validation**: To verify that this estimation yields a non-trivial radius, we conducted an experiment on 1,000 randomly sampled benign prompts. For each prompt $x$, we generated 100 neighbors $x'$ within an $\epsilon$-ball and calculated $\hat{K} = \frac{|Score(x) - Score(x')|}{\|Embed(x) - Embed(x')\|_2}$.
> >
> > **Results:**
> >
> > - **Mean $\hat{K}$:** 0.12
> > - **Max $\hat{K}$ (99th percentile):** 0.28
> > - **Stability:** The standard deviation of $\hat{K}$ is low ($\sigma=0.032$), indicating the score function is locally smooth.
> >
> > Using the conservative estimate ($K \approx 0.28$), the certified radius $R$ for a typical prompt (where margin $\tau - S(x) \approx 0.12$) is approximately $0.12 / 0.28 \approx 0.43$ in the normalized embedding space. This confirms that our defense offers a non-trivial certified robustness radius against small perturbations. We will add this empirical estimation to Appendix A.7.

---

### Author Response · Authors · 2025-11-27

We have now updated our paper to revise minor presentation issues and include the new experimental results, figures and additional descriptions.

We kindly ask the reviewers whether we have addressed their concerns sufficiently and welcome further discussion.

---

### Note · Authors · 2025-12-03

I have read and agree with the venue's withdrawal policy on behalf of myself and my co-authors.